# Temporal Skin Temperature as an Indicator of Cardiorespiratory Fitness Assessed with Selected Methods

**DOI:** 10.3390/biology11070948

**Published:** 2022-06-21

**Authors:** Agnieszka Danuta Jastrzębska, Rafal Hebisz, Paulina Hebisz

**Affiliations:** Department of Physiology and Biochemistry, Wroclaw University of Health and Sport Sciences, 51-612 Wroclaw, Poland; rafal.hebisz@awf.wroc.pl (R.H.); paulina.hebisz@awf.wroc.pl (P.H.)

**Keywords:** cardiovascular fitness, aerobic capacity, skin surface temperature, high-intensity exercise, thermal imaging, recovery

## Abstract

**Simple Summary:**

The aim of this study was to investigate whether it is possible to use infrared thermography to assess cardiovascular fitness and aerobic capacity. Changes in temporal temperature during and after a single bout of high-intensity exercise were measured from subjects with varying levels of physical activity. Significant correlation between the temporal temperature measured during recovery time with cardiovascular fitness parameters (HRR and HRV) and maximum oxygen consumption confirm the usefulness of thermal imagining in aerobic capacity evaluation. These results could foster the employment of infrared thermography to monitor athletic/athletes’ performance.

**Abstract:**

The aim of this study was to determine whether there are associations between cardiovascular fitness (and aerobic capacity) and changes in temporal skin temperature during and after a single bout of high-intensity exercise. Twenty-three men with varying levels of physical activity (VO_2_max: 59.03 ± 11.19 (mL/kg/min), body mass 71.5 ± 10.4 (kg), body height 179 ± 8 (cm)) participated in the study. Each subject performed an incremental test and, after a 48-h interval, a 110%Pmax power test combined with an analysis of the thermal parameters, heart rate recovery and heart rate variability. Thermal radiation density from the body surface (temple) was measured using a Sonel KT384 thermal imaging camera immediately after warm-up (Tb), immediately after exercise (Te) and 120 sec after the end of exercise (Tr). The differences between measurements were then calculated. The correlation analysis between the thermal and cardiovascular function parameters during the recovery period showed strong positive associations between the Tr-Te difference and measures of cardiovascular fitness (50 < r < 69, *p* < 0.05). For example, the correlation coefficient between Tr-Te and VO_2_max reached 0.55 and between Tr-Te and Pmax reached 0.68. The results obtained indicate that the measurement of temporal temperature during and after an intense 3-min bout of exercise can be used to assess aerobic physical capacity and cardiovascular fitness.

## 1. Introduction

The evaluation of cardiovascular fitness during exercise has been the subject of numerous research papers [1,2,3,4,5,6,7,8,9]. In exercise testing, the measurement of maximal oxygen uptake (VO_2_max) [1,2,3,4], assessment of heart rate recovery (HRR) [5,6,7] and heart rate variability (HRV) [8,9] are popular methods used in its assessment.

The efficiency of the cardiovascular system (especially the cardiac minute volume) and the blood volume [10] influence the regulation of blood distribution to the muscles and skin during exercise [11]. Regulation of blood flow is largely an adrenergic response. During intense exercise, the concentration of norepinephrine in blood increases [12,13]. Norepinephrine acts on the postsynaptic receptors (alpha1 and alpha2) of the sympathetic adrenergic system, causing vasoconstriction of the cutaneous circulation vessels [11,14]. As a result, at the onset of intense exercise, there are changes in blood flow, consisting of an increase in flow in the limbs loaded with exercise and a decrease in flow to inactive limbs and skin microcirculation [14,15,16]. Through the described reactions, the muscles are better supplied with oxygen during intense work and the convection of flowing blood is more effective [16,17,18,19,20,21]. However, when exercise is performed over a long period of time, the internal body temperature increases and the heat removal mechanisms must be activated [22]. These include the vasodilation of the skin vessels and an increase in the intensity of cutaneous blood flow, intended to increase the heat release to the atmosphere [22]. The magnitude of the increase in cutaneous blood flow during exercise depends on aerobic capacity [11] and increased adaptation to training and heat stress [23]. This adaptation lowers the internal temperature threshold at which vasodilation of the skin vessels occurs [23]. Exercise-induced changes in blood flow affect body surface temperature [16,24].

The previous study by Hebisz et al. (2019) [25] showed that the level of maximal oxygen uptake correlates strongly with the change in the temporal temperature observed during recovery. At the same time, no similar correlations (with maximal oxygen uptake) by measuring exercise and recovery changes in arm temperature were established. It is surmised that those effects may be related to the degree of the vascularisation of the inspected skin region, which is large in the temporal area (compared with the vascularisation of the arm), due to a branch of the external carotid artery (superficial temporal artery) [26].

The exercise protocol used in the aforementioned study consisted of 4 sprints of 30 s each, separated by 90 s rest periods [25]. Repeated sprint efforts (similar to Wingate tests) lead to a severe disturbance of acid-base homeostasis [27] and high levels of subjective fatigue [28]. For these reasons, there are people who avoid such efforts [28]. An alternative protocol for testing thermal parameters on the temporal surface could be based on a single effort of a few minutes, typical for HIIT-type training or tests aimed at verifying maximal oxygen uptake. Such efforts are also very intensive as they achieve an oxygen uptake close to the maximum [29], but at the same time, the level of subjective fatigue may be lower afterwards [28].

The aim of this study was to determine whether there are associations between cardiovascular fitness (and aerobic capacity) and changes in temporal skin temperature during and after a single bout of high-intensity exercise. It was assumed that, among other things, the rate of maximal oxygen uptake (as well as the rate of heart rate recovery) would positively correlate with the change in temporal temperature post-exercise.

## 2. Material and Methods

### 2.1. Participants

A group of 23 men with varying levels of physical activity participated in the study. The participants were free of any known neuromusculoskeletal, cardiovascular and respiratory systems impairment. A total of 6 participants led a sedentary lifestyle, 10 were classified as a physically active (exercise duration 3 to 5 h per week: 5—runners, 3—cross-fit, 2—swimming, 1—racket games;) and 7 were classified as athletes (regularly participating in sport competitions; exercise duration 9 to 15 h per week: 3—runners, 2—cross-country skiing, 2—team games). Table 1 shows the values of parameters characterising the physical capacity and physique of the study group.

The study was approved by the University Ethics Committee and conducted in accordance with the ethical standards established by the Declaration of Helsinki. Participants were made aware of the experimental protocol and gave written consent to participate prior to the study.

### 2.2. Study Design

The subjects had not performed heavy physical exercise in the 48 h prior to the exercise tests. Each subject performed an incremental test for cardiovascular fitness and aerobic efficiency assessment as well to determine the power of the verification test equal to 110% of Pmax obtained in GXT. After a 48-h interval, a 110%Pmax power test combined with an analysis of the thermal parameters, heart rate recovery and heart rate variability were carried out (Figure 1).

#### 2.2.1. Body Composition

Before the test, body composition was measured using a near-infrared device NIR (6100/XL, Futrex, Hagerstown, MD, USA). The device measures the optical density of body tissues on the *biceps brachii* of the dominant hand to estimate the body fat expressed as kilograms and percentages of body weight, lean body mass (LBM) and body water content in liters and as a percentage of the body mass. It is a commonly used method for body composition measurements [30,31].

#### 2.2.2. Graded Exercise Test (GXT)

A graded exercise test (GXT) was performed to determine aerobic power [25,32]. The test was conducted on a Lode Excalibur Sport cycloergometer (Lode B.V., Groningen, The Netherlands), which was calibrated before the start of the study. The seat height was adjusted individually so that the angle of deflection was no greater than 5° when the leg was fully extended. The effort started at a load of 50 W; every 3 min, the load was increased by 50 W until the subject refused to continue. If during the last load of the test the subject failed to exert for 3 min, then for each missing second, 0.28 W was subtracted from the current power value [33]. In this way, the maximal aerobic power (Pmax) value was obtained.

During testing, subjects breathed through a mask as their expired air was sampled breath by breath and analysed by a Quark CPET gas analyser (Cosmed, Milan, Italy). The apparatus was calibrated with atmospheric air and a gas mixture composed of the following elements: CO_2_—5%, O_2_—16% and N_2_—79%. The respiratory parameters (oxygen uptake (VO_2_), exhaled carbon dioxide (VCO_2_), and minute pulmonary ventilation (VE)) were measured. The analysis of the data was carried out with the results averaged every 30 s. The highest VO_2_ recorded in GXT defines VO_2_peak1. Heart rate (HR) was recorded with the V800 cardiofrequencimeter (Polar, Oy, Kempele, Finland).

The first ventilatory threshold (VT1), at the point before the first non-linear increase in VE·VO_2_^−1^ equivalent without a concomitant increase in VE·VCO_2_^−1^, was determined from the recorded respiratory data from the GXT test; the second ventilation threshold (VT2) was determined at the point preceding the second non-linear increase in VE·VO_2_^−1^ or VE·VCO_2_^−1^ equivalent, following the methodology described by Beaver et al. (1986) [34] and Davis et al. (1980) [35].

#### 2.2.3. Test at 110% of Pmax

A Lode Excalibur Sport cycloergometer and a Quark ergospirometer were used to perform the test. The test was preceded by a 15-min warm-up, consisting of exercising for 5 min at an intensity corresponding to the power achieved at the VT1, followed by 10 min at a power corresponding to halfway between the VT1 and the VT2. The warm-up was followed by a 10-min passive break. The verification test lasted 3 min and was performed at an intensity of 110% of the Pmax achieved in the GXT test performed two days earlier. Recording of the respiratory parameters began 1 min before the verification test and ended 5 min after the test. The analysis of the data was carried out using the results averaged every 30 s. The highest recorded oxygen uptake value (from 30 sec averaging) was considered to be the peak oxygen uptake in a verification test performed on a separate day (VO_2_peak2). Maximal oxygen uptake (VO_2_max) was considered to be the higher than VO_2_peak1 and VO_2_peak2, as in earlier work [32]. The power and oxygen uptake parameters analysed in this study were recalculated in relation to body mass (VO_2_peak1, VO_2_peak2 and VO_2_max) and lean body mass (Pmax·LBM^−1^, VO_2_max·LBM^−1^). Each exercise test was performed under controlled thermal conditions. The air temperature was 21 °C and the humidity was between 40–45%.

During the test at power 110%Pmax, the density of thermal radiation from the body surface was measured using a thermal imaging camera Sonel KT384 (Sonel S.A., Świdnica, Poland). The thermal imaging camera was used in accordance with the manufacturer’s guidelines. The camera features IR resolution of 384 × 288; spectral range of 8–14 μm; thermal sensitivity of 0.08 °C. The software provided by the manufacturer (Sonel ThermoAnalyze ver. 1.0) converted radiation density into body surface temperature expressed in °C. During playback of the recorded video, individual frames were analysed and the mean temperature was recorded within a square box (with a side length of 10 pixels), individually marked on the temple (Figure 2), as in an earlier publication [25]. Baseline temperature was determined after warm-up (T_b_), immediately before the start of the exercise at 110%Pmax. Further temperature determinations were made immediately after exercise (T_e_), and 120 s after exercise (T_r_). The difference between T_e_ and T_b_ (T1 = T_e_ − T_b_), the difference between T_r_ and T_b_ (T2 = T_r_ − T_b_) and the difference between T_r_ and T_e_ (T3 = T_r_ − T_e_) were then calculated.

During the warm-up and a 10-min passive break before the 110%Pmax power test, the time interval between heartbeats (RR) was recorded using a V800 cardiofrequency meter (Polar, Oy, Kempele, Finland). Heart rate values were averaged for 59–61” (HRR1’), 119–121” (HRR2’), 179–181” (HRR3’), 239–241” (HRR4’) and 299–301” (HRR5’) recovery after warm-up, as recovery RR interval measurements have a high variability. The RR intervals allowed for the analysis of consciously selected sections for HRR analysis. A simpler method, such as HR recording, would result in a random entry segment for the HRR analysis, as the cardiofrequencymeter’s software calculates the HR based on the averaging data at 3 s intervals. The conversion of RR to HR was performed automatically by PolarFlow (www.flow.polar.com accessed on 22 October 2021) in a precisely marked part of the saved data. The changes in heart rate during recovery after warm-up were then calculated as the difference between the heart rate measured at the end of warm-up and HRR1’ (ΔHRR1’), HHR2’ (ΔHRR2’), HRR3’ (ΔHRR3’), HRR4 (ΔHRR4’) and HRR5’ (ΔHRR5’), respectively, similar to Suzic Lazic et al. (2017) [6].

For the temporal HRV parameters, the square-root of the mean squared difference between successive normal-to-normal RR intervals (RMSSD_3–5′_) and standard deviation of normal-to-normal RR intervals (SDNN_3–5′_) were calculated from the data recorded by the cardiofrequencimeter during recovery after the warm-up. For the frequency domain, a spectral analysis was performed using fast Fourier transformation to obtain low-frequency spectral power (LFP_3–5′_) and high-frequency spectral power (HFP_3–5′_). For this purpose, the Kubios HRV Standard software (KubiosOy, Kuopio, Finland) was used. HRV parameters were calculated for the portion of the recording covering the third, fourth and fifth minutes of recovery, similar to the methodology previously used by Buchheit et al. (2009) [36]. Medium threshold data filtering was applied in the calculation of the above variables.

For the analyses of HRV, HRR and ΔHRR, we used data collected in post-warm-up recovery as these data allow analyses to be performed after standardised moderate-intensity exercise, similar to the procedures used by other authors [36,37].

### 2.3. Statistical Analysis

The Shapiro–Wilk test was used to assess the distribution of the parameters studied. Analysis of variance with repeated measurements was used to compare the parameters that were measured several times during the study. Pearson’s simple correlation was used to determine the strength of the relationships between the changes in temporal temperature and measurements of cardiovascular fitness (and physical capacity). The scale modified by Hopkins et al. (2009) [38] was used to interpret the correlation coefficient, which is as follows: 0.1–0.29 = trivial, 0.30–0.49 = moderate, 0.50–0.69 = strong, 0.70 to 0.89 = very strong, 0.90–0.99 = nearly perfect, and 1 = perfect.

Using the formula for the critical value of the correlation coefficient, the minimum group size was calculated assuming that the acceptable level of statistical significance (alpha) is 0.05 and that the strength of the correlation should be high, *r* > 0.5. The following formula was used:(1)$$r=\sqrt{\frac{t_{\alpha }^{2}}{n-2+t_{\alpha }^{2}}}\;$$

On this basis, we determined that the minimum number of study participants was 16 at *t* ≈ 2.15.

## 3. Results

Using the analysis of variance, a statistically significant effect of the repeated measurements was demonstrated for Temp_b,e,r_ (F = 39.95; *p* = 0.000; η^2^ = 0.645). Using a post-hoc test, significant differences were shown between Temp_e_ and Temp_b_ and between Temp_r_ and Temp_e_ (Table 2).

The simple correlation coefficient indicated moderate statistically significant relationships of T1 with VO_2_peak2, VO_2_max, VO_2_max·LBM^−1^ (0.30 < r < 0.49) and strong relationships of T3 with Pmax, Pmax·LBM^−1^, VO_2_peak1, VO_2_peak2, VO_2_max, VO_2_max·LBM^−1^ (Table 3) (0.50 < r < 0.69).

Using the analysis of variance, statistically significant effects of the repeated measurements were demonstrated for HRR (F = 15.24; *p* = 0.000; η^2^ = 0.409) and ΔHRR (F = 15.57; *p* = 0.000; η^2^ = 0.414). Using post-hoc tests, it was shown that HRR and ΔHRR measurements performed in the first minute of recovery were significantly different from those performed in the second, third, fourth and fifth minutes of recovery (Table 4).

Using the simple correlation coefficient, strong (0.50 < *r* < 0.69), statistically significant relationships of T1 with ΔHRR2’, ΔHRR3’ and T3 with HRR1’, HRR2’ HRR4’, HRR5’, ΔHRR1’ ΔHRR2’ ΔHRR4’ ΔHRR5’ were demonstrated (Table 5).

Using the simple correlation coefficient, moderate (0.30 < *r* < 0.49), statistically significant associations (*p* < 0.05) of T2 with SDNN1 and strong (0.50 < *r* < 0.69) of T3 with all HRV parameters analysed in recovery were demonstrated (Table 6).

## 4. Discussion

The results of the study presented in this paper confirm the previous findings that during a few minutes of intense physical exercise, the body surface temperature decreases [24,39,40], which is linked to vasoconstriction of the vessels of cutaneous circulation and consequently to a decrease in blood flow in cutaneous circulation [24,39]. Furthermore, the results presented herein indicate that the level of cardiovascular fitness is related to the magnitude of the decrease in temporal temperature during an effort of 110%Pmax lasting a few minutes. This is evidenced by the results of the Pearson analyses, which showed a negative correlation of moderate strength between T1 and VO_2_max and ΔHRR2’ and ΔHRR3’. The presence of the above correlations may be due to the fact that individuals with higher VO_2_max [41] and greater HRR [42] are able to achieve greater power output in incremental tests and, therefore, also in the 110%Pmax intensity test. During the initial phase of high-power exercise, oxygen deficit and muscle oxygen demand increase rapidly [43], resulting in vasoconstriction in the inactive tissues (including the skin) and redirection of blood towards the active tissues [21]. Hence, the magnitude of the decrease in exercise body surface temperature is dependent on the intensity of the exercise [24]. The mechanism described above may also explain the correlations that relate T1 and VO_2_max, as well as T1 (Table 3) and ΔHRR, in our study (Table 2). Thus, the magnitude of the drop in temporal temperature can be used as an indicator of the level of cardiovascular fitness alongside generally accepted indicators such as maximal oxygen uptake [44] and heart rate recovery [7,8,9].

The mechanical efficiency of muscles is only 20–30% [45,46]. When exercise is continued over a long period of time, the internal body temperature increases [47] because the energy created by the skeletal muscle metabolism is largely converted into thermal energy [48]. Maintaining a balance between heat production and heat removal during exercise ensures that you can continue exercising, preventing the development of thermal shock [49]. Thermal energy is most effectively removed from the muscles by blood convection [50]. At the same time, vasodilatation of the vessels of cutaneous circulation takes place, which enables an increase in cutaneous blood flow, an increase in body surface temperature and the release of thermal energy through radiation and sweat evaporation [21]. In the process of training, vascular endothelial function improves [16]. This has been demonstrated by administering acetylcholine and sodium nitroprusside by iontophoresis [51,52]. In conjunction with improved vascular endothelial function, it was observed that endurance-trained individuals had higher cutaneous blood flow during exercise at intensities up to 90% VO_2_max, compared to non-trained individuals [53]. In addition, endurance-trained athletes have a higher blood volume and a higher cardiac stroke volume than non-trained athletes [54,55]. High cutaneous blood flow, high cardiac stroke volume and high plasma volume are factors that determine VO_2_max levels [2,55] and enable efficient thermal energy removal during exercise [56,57]. Therefore, it can be assumed that high cardiovascular fitness should favour a higher increase in body surface temperature, which is confirmed by the strong correlation between T3 and VO_2_max (0.50 < r < 0.69) (Table 3). The direction of the correlation of T3 with VO_2_max is opposite to the correlation between T1 and VO_2_max probably because for short high-intensity efforts, the increase in body surface temperature occurs only after the end of the effort [58].

The relationship between T3 and VO_2_max (Table 3) that we observed in recovery is similar to the observations described in our earlier publication [25], in which we showed a correlation between VO_2_max and recovery temporal temperature change after sprint interval training (4 all-out sprints of 30 s each, interspersed with breaks of 90 s). However, the exercise test protocol used in previous studies [25] is not commonly used in the diagnosis of performance capacity. Efforts lasting several minutes at 110% Pmax are more commonly performed in tests to verify VO_2_max [32,59,60,61]. Similar efforts are also used during training [29]. For this reason, the correlation described in this study, with an exercise protocol lasting 3 min at 110%Pmax, seems to be of greater applicative importance.

In the current study, HRV recovery and ΔHRR were measured after a moderate-intensity warm-up, and T3 was measured during the high-intensity exercise that followed the warm-up. Nevertheless, our results show strong correlations of T3 with ΔHRR and with HRV (Table 6) parameters in each of the five recovery minutes analysed (Table 5). As mentioned above, HRR is a measure of cardiovascular fitness [8,9] as well as fitness level in endurance disciplines [6]. During recovery, the heart rate is reduced by a decrease in sympathetic, and an increase in parasympathetic, nervous system activity [6,62]. HRV parameters measured during recovery are also a measure of sympathovagal balance [8]. This balance depends, among other things, on the level of heat stress [63,64]. It is, therefore, possible that T3 correlates with ΔHRR and HRV recovery because it is a measure of the ability to reduce heat stress. The significance of this relationship appears to be high, as the coefficient of determination (R^2^) between T3 and recovery HRV parameters reached 45% in our analyses (T3 and RMSSD_3–5′_).

Maximal oxygen uptake is the parameter that determines the possible amount of energy obtained through aerobic metabolism [65]. In incremental tests, aerobic metabolism is responsible for the vast majority of the work carried out and even in the final phase of these tests, it is the dominant energy source [66]. Moreover, the correlation between maximal power in incremental tests and VO_2_max level is very strong [67]. In view of this, the relationship of T3 and VO_2_max should allow a correlation of T3 with Pmax to exist and the current results shows this correlation, which was strongest (r = 0.68) when Pmax was expressed as Pmax·LBM^−1^ (Table 3). This is a significant outcome of the current study, as it allows us to conclude that physical capacity (ability to perform intense aerobic work) can be assessed on the basis of changes in post-exercise body surface temperature.

The results of the research presented in this paper indicate that the temperature of the temples’ surface increases in recovery after intense aerobic exercise, especially in people with high physical capacity. Such results confirm the thesis from the studies by Hebisz et al. [25] that blood flow increases in the temporal region as a result of intensive work. At the same time, already at an intensity above 60% VO_2_max, the cerebral blood flow decreases, which protects the brain from excessive thermal stress [68]. It is possible that the reduction in cerebral blood flow additionally increases the blood flow in the branches of the external carotid artery. Thus, a decrease in cerebral blood flow may result in a greater increase in the temples’ surface temperature following exercise. However, this statement requires empirical confirmation.

## 5. Conclusions

As hypothesised, the recovery changes in temporal temperature (after a 3-min bout of intense exercise) were positively correlated with the measurements of cardiovascular fitness applied. It is also possible to assess the level of aerobic capacity (ability to perform intense aerobic work) on the basis of recovery changes in temporal temperature. Furthermore, exercise-induced changes in temporal temperature correlated with the measurements of cardiovascular fitness, although in this case, the relationship was negative. The results obtained indicate that the measurements of temporal temperature during and after an intense 3-min bout of exercise can be used to assess aerobic physical capacity and cardiovascular fitness.

## Figures and Tables

**Figure 1 biology-11-00948-f001:**
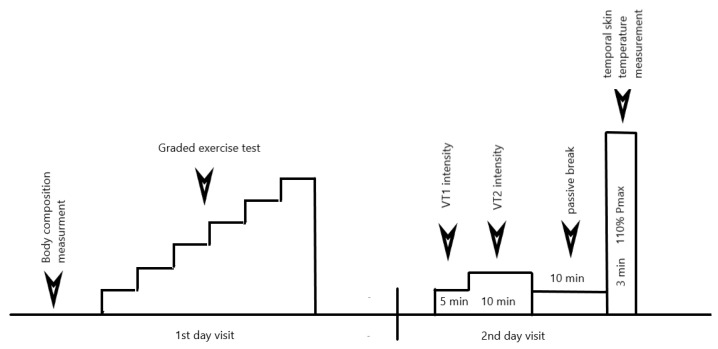
Scheme of visit in laboratory.

**Figure 2 biology-11-00948-f002:**
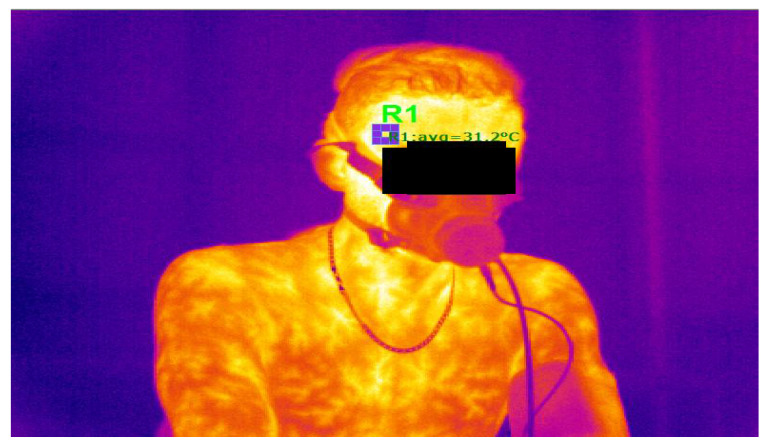
Single frame of film recorded during a test at 110%Pmax in which the field (R1) for temporal surface temperature analysis is marked.

**Table 1 biology-11-00948-t001:** Physiological and anthropometric characteristics of the group.

VO_2_max	Pmax	Age (y)	BM (kg)	LBM (kg)	BH (cm)
59.03 ± 11.19	332.5 ± 48.4	22.8 ± 5.4	71.5 ± 10.4	64.3 ± 8.6	179 ± 8.1

VO_2_max—maximum oxygen consumption; Pmax—maximum power in graded exercise test; BM—body mass; LBM—lean body mass; BH—body height.

**Table 2 biology-11-00948-t002:** Thermal parameters and baseline measurements of the physical capacity of the study participants.

Items	X ± SD	Lower 95%CI	Upper 95%CI
Temp_b_ (°C)	33.0 ± 2.5	31.9	34.0
Temp_e_ (°C)	30.8 ± 2.5 *	29.7	31.9
Temp_r_ (°C)	32.6 ± 2.6 **	31.5	33.7
T1 (°C)	−2.2 ± 1.1	−2.7	−1.7
T2 (°C)	−0.6 ± 1.3	−1.2	0.0
T3 (°C)	1.7 ± 1.0	1.3	2.1
Pmax (W)	332.5 ± 48.4	311.6	353.5
Pmax·LBM^−1^ (W·kg^−1^)	5.27 ± 1.04	4.82	5.72
VO_2_peak1 (mL·min^−1^·kg^−1^)	55.56 ± 13.02	49.93	61.19
VO_2_peak2 (mL·min^−1^·kg^−1^)	58.25 ± 10.82	53.57	62.93
VO_2_max (mL·min^−1^·kg^−1^)	59.03 ± 11.19	54.19	63.87
VO_2_max·LBM^−1^ (mL·min^−1^·kg^−1^)	65.35 ± 11.21	60.50	70.20

Temp_b_—baseline temporal temperature; Temp_e_—temporal temperature measured immediately after the exercise at 110%Pmax; Temp_r_—temporal temperature measured during recovery—120 sec after the exercise at 110%Pmax; T1—difference between Temp_e_ and Temp_b_; T2—difference between Temp_r_ and Temp_b_; T3—difference between Temp_r_ and Temp_e_; Pmax—maximum aerobic power; LBM—lean body mass; VO_2_peak1—peak oxygen uptake determined in GXT test; VO_2_peak2—peak oxygen uptake determined in exercise with 110%Pmax; VO_2_max—maximum oxygen uptake; CI—confidence interval; *—*p* < 0.000 vs. Temp_b_; **—*p* < 0.000 vs. Temp_e_.

**Table 3 biology-11-00948-t003:** Pearson correlation between thermal parameters and baseline measurements of physical capacity.

Items	T1 (°C)	T2 (°C)	T3 (°C)
Pmax (W)	−0.33	0.11	0.63 *
Pmax·LBM^−1^ (W·kg^−1^)	−0.32	0.06	0.68 *
VO_2_peak1 (mL·min^−1^·kg^−1^)	−0.37	−0.07	0.46 *
VO_2_peak2 (mL·min^−1^·kg^−1^)	−0.42 *	−0.01	0.51 *
VO_2_max (mL·min^−1^·kg^−1^)	−0.43 *	−0.05	0.51 *
VO_2_max·LBM^−1^ (mL·min^−1^·kg^−1^)	−0.42 *	−0.02	0.55 *

T1—difference between Temp_e_ and Temp_b_; T2—difference between Temp_r_ and Temp_b_; T3—difference between Temp_r_ and Temp_e_; Pmax—maximum aerobic power; LBM—lean body mass; VO_2_peak1—peak oxygen uptake determined in GXT test; VO_2_peak2—peak oxygen uptake determined in exercise with 110%Pmax; VO_2_max—maximum oxygen uptake; *—*p* < 0.05.

**Table 4 biology-11-00948-t004:** Recovery heart rate and recovery heart rate variability among study participants.

Items	1′	2′	3′	4′	5′
HRR (bpm) 95%CI L–U	99.2 ± 17.1 91.8–106.6	90.5 ± 18.6 * 43.7–54.6	88.4 ± 16.8 * 81.2–95.7	86.7 ± 14.8 * 80.3–93.1	86.2 ± 12.5 * 80.8–91.6
ΔHRR (bpm) 95%CI L–U	49.1 ± 12.6 43.7–54.6	58.1 ± 13.9 * 52.1–64.2	60.2 ± 13.3 * 54.4–65.9	62.0 ± 11.2 * 57.1–66.8	62.4 ± 11.0 * 57.7–67.2
SDNN_3–5′_ (ms) 95%CI L–U	---	---	43.10 ± 22.46 33.39–52.81		
RMSSD_3–5′_ (ms) 95%CI L–U	---	---	31.17 ± 19.72 22.65–39.70		
HFP_3–5′_ (ms^2^) 95%CI L–U	---	---	657.1 ± 1005.7 222.2–1092.0		
LFP_3–5′_ (ms^2^) 95%CI L–U	---	---	1488.2 ± 1541.9 821.5–2155.0		

HRR—heart rate recovery; ΔHRR—difference between heart rate measured at the end of the warm-up and the heart rate measured at the end if 1′, 2′, 3′, 4′, 5′ minute of recovery; SDNN_3–5′_—the standard deviation of NN intervals; RMSSD_3–5′_—the root mean square of successive differences between normal heart beats HFP_3–5′_—high-frequency power; LFP_3–5′_—low-frequency power; CI L–U—confidence interval for lower – upper value; 1′, 2′ 3′, 4′, 5′—1′, 2′ 3′, 4′, 5′—measurements taken in the next minutes of recovery after the warm-up; *—*p* < 0.05 vs. 1′.

**Table 5 biology-11-00948-t005:** Pearson correlation between thermal parameters and recovery heart rate.

Items	T1 (°C)	T2 (°C)	T3 (°C)
HRR1′ (bpm)	0.09	−0.28	−0.53 *
ΔHRR1′ (bpm)	−0.34	0.11	0.63 *
HRR2′ (bpm)	0.20	−0.18	−0.50 *
ΔHRR2′ (bpm)	−0.48 *	−0.02	0.57 *
HRR3′ (bpm)	0.17	−0.16	−0.36
ΔHRR3′ (bpm)	−0.44 *	−0.07	0.35
HRR4′ (bpm)	0.10	−0.3	−0.58 *
ΔHRR4′ (bpm)	−0.40	0.07	0.64 *
HRR5′ (bpm)	0.12	−0.48 *	−0.62 *
ΔHRR5′ (bpm)	−0.41	0.22	0.57 *

T1—difference between Temp_e_ and Temp_b_; T2—difference Temp_r_ and Temp_b_; T3—difference Temp_r_ and Temp_e_; HRR—heart rate recovery; ΔHRR—difference between heart rate measured at the end of the warm-up and the heart rate measured at the end if 1′, 2′, 3′, 4′, 5′ minute of recovery; 1′, 2′, 3′, 4′, 5′—measurements taken in the next minutes of recovery after the warm-up; *—*p* < 0.05.

**Table 6 biology-11-00948-t006:** Pearson correlation between thermal parameters and recovery heart rate variability parameters.

Items	T1 (°C)	T2 (°C)	T3 (°C)
SDNN_3–5′_	−0.18	0.43 *	0.65 *
RMSSD_3–5′_	−0.29	0.36	0.67 *
HFP_3–5′_	−0.22	0.29	0.53 *
LFP_3–5′_	−0.17	0.31	0.50 *

T1—difference between Temp_e_ and Temp_b_; T2—difference between Temp_r_ and Temp_b_; T3—difference between Temp_r_ and Temp_e_; SDNN—the standard deviation of NN intervals; RMSSD—the root mean square of successive differences between normal heart beats HFP—high-frequency power; LFP—low-frequency power; 1′, 2′ 3′, 4′, 5′—measurements taken in the next minutes of recovery after the warm-up; *—*p* < 0.05.

## Data Availability

The data are available on request from the corresponding author.

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
