# Peer review of "Temporal Skin Temperature as an Indicator of Cardiorespiratory Fitness Assessed with Selected Methods"

_biology, 2022, doi:10.3390/biology11070948_

Round 1

Reviewer 1 Report

Important revisions to be made regarding formatting errors and comments specific to the different sections of the article are below:

Important formatting mistakes

-Line 14: athletes´ performance.

-Line 227: Note Table 5, replace warm up instead of worm up*,

-General comment: authors should remember that scientific articles are not written in the first person, therefore all statements that include "we" (in our previous study, our previous results...etc.) should be transformed to impersonal voice.

Specific comments

Abstract

-Line 16: “changes in temporal skin temperature”

-Line 30: “temporal skin temperature” is in the title, therefore it cannot be in the keywords and should be replaced.

Introduction

-Line 35: write down the references of the numerous research papers.

-Line 52: which reference supports this information? In this case, is “the core temperature” or “the body temperature”?

-Lines 62-64: The introduction has to justify everything that is later developed in the discussion. However, it is not necessary to focus on the results of previous studies. Moreover, it should not be used in the first person “we did not establish”. It is better “it was not established…”

Material and methods

2.1. Participants

 -Line 88: Much more information is required from the participants, e.g. specifying whether the participants were physically active, whether they were healthy or had any illnesses, whether they had experience of sport (this would be important in the case of cycling).

Study design

-Lines 96-99: this paragraph should be placed in the “study design” instead of “participants” section.

-Line 102: checking the parenthesis within a parenthesis

Body composition

-Line 100: this section from line 100 to line 105 should be placed in a paragraph with the heading "body composition" where again more information should be provided on the measurements collected through the NIR device. For example, perimeters. In this case, as a reviewer, I advise you to provide all those measurements that can be measured with this device as descriptive information, even if they are not analysed in the article.

Graded Exercise Test

-Lines 107-115: Why has a previously validated protocol not been selected? In case this protocol has been validated or used before, it should be reflected in some references. The GXT protocol has been extensively studied in the literature, therefore, this protocol should be supported by the literature.

For example: Javaloyes, A., Sarabia, J. M., Lamberts, R. P., Plews, D., & Moya-Ramon, M. (2020). Heart rate variability-guided training prescription versus block periodization in well-trained cyclists. The Journal of Strength & Conditioning Research, 34(6), 1511-1518.

Lines 112-113: can the authors support this formula in the literature?

Author Response

Response to Reviewer 1 Comments

General comment for Reviewer 1: Thank you for the review comments. We read it in detail and we were trying to make all improvements with the best effort.

As the process of changing the original text was complex, all inserted new text, parts of the text, new words or parts of words were marked in red letters. Corrections are marked in light blue, as suggested by the reviewer. In green and yellow, corrections as suggested by other reviewers.

Important formatting mistakes

-Line 14: athletes´ performance.

Response 1. We agree with the suggestion and it was corrected.

-Line 227: Note Table 5, replace warm up instead of worm up*,

Response 2. Thank you for this comment. That was corrected throughout the text.

-General comment: authors should remember that scientific articles are not written in the first person, therefore all statements that include "we" (in our previous study, our previous results...etc.) should be transformed to impersonal voice.

Response 3. We fully agree with the reviewer, and we do all changes according to suggestions.

Specific comments

Abstract

-Line 16: “changes in temporal skin temperature”

Response 4. Accordingly, to suggestion we changed “changes in temporal temperature” to “changes in temporal skin temperature”.

-Line 30: “temporal skin temperature” is in the title, therefore it cannot be in the keywords and should be replaced.

Response 5. The temporal skin temperature was replaced by skin surface temperature

Introduction

-Line 35: write down the references of the numerous research papers.

Response 6. The references were supplemented.

-Line 52: which reference supports this information? In this case, is “the core temperature” or “the body temperature”?

Response 7. The core temperature was replaced by internal body temperature and the relevant reference was supplemented.

-Lines 62-64: The introduction has to justify everything that is later developed in the discussion. However, it is not necessary to focus on the results of previous studies. Moreover, it should not be used in the first person “we did not establish”. It is better “it was not established…”

Response 8. The sentence  „Taking this into account, in our own earlier work we hypothesised that changes in body surface temperature during high-intensity exercise might be related to the level of aerobic physical capacity and the level of cardiovascular fitness [25]” has been removed, and the paragraph was redrafted so as not to refer directly to the results of previous studies. Also, the impersonal voice was used.

Material and methods

2.1. Participants

 -Line 88: Much more information is required from the participants, e.g. specifying whether the participants were physically active, whether they were healthy or had any illnesses, whether they had experience of sport (this would be important in the case of cycling).

Response 9. More detailed information was added with the Reviewer's suggestion “The participants were free of any known neuromusculoskeletal, cardiovascular and respiratory systems impairment. Six of the participants led a sedentary lifestyle, 10 were classified as a physically active (exercise duration 3 to 5 hours per week: 5 – runners, 3 – cross-fit, 2 - swimming, 1 – rocket games;), 7 were classified as athletes (regularly participating in sport competitions; exercise duration 9 to 15 hours per week: 3 – runners, 2 – cross-country skiing, 2 – team games”

Study design

-Lines 96-99: this paragraph should be placed in the “study design” instead of “participants” section.

Response 10. With the Reviewer's suggestion, this paragraph was replaced to “study design”

-Line 102: checking the parenthesis within a parenthesis

Response 11. The parenthesis was removed.

Body composition

-Line 100: this section from line 100 to line 105 should be placed in a paragraph with the heading "body composition" where again more information should be provided on the measurements collected through the NIR device. For example, perimeters. In this case, as a reviewer, I advise you to provide all those measurements that can be measured with this device as descriptive information, even if they are not analysed in the article.

Response 12. We agree with the reviewer's comment. We added the heading „Body composition”, we provided also other parameters that can be measured using Futrex analyser.

“The device measures the optical density of body tissues on the biceps brachii of the dominant hand to estimate the body fat expressed as kilograms and percentages of body weight,  lean body mass (LBM) and body water content in liters and as a percentage of the body mass..”

Graded Exercise Test

-Lines 107-115: Why has a previously validated protocol not been selected? In case this protocol has been validated or used before, it should be reflected in some references. The GXT protocol has been extensively studied in the literature, therefore, this protocol should be supported by the literature.

Response 13. In the presented research work, we used the protocol that we described in previous publications. This protocol has been used by us for a long time and we do not want to change it so that we can obtain comparable data to previous studies. We added appropriate citation in the description of "Graded exercise test"

Lines 112-113: can the authors support this formula in the literature?

Response 14. This formula was used following the:

Jeukendrup A, Saris WH, Brouns F, Kester AD. A new validated endurance performance test. Med Sci Sports Exerc. 1996 Feb;28(2):266-70. doi: 10.1097/00005768-199602000-00017. PMID: 8775164.

08.06.2022       Corresponding author

Reviewer 2 Report

In the current manuscript, the Authors aimed to investigate whether it is possible to use infrared 8 thermography in the assessment of cardiovascular fitness and aerobic capacity. Although the current issue is interesting, some points are addressed to the Authors, and they need be clarified to improve the manuscript’s quality.

Considering Abstract, in general, this section is well written. Some few points need to be improved; in Results, please, consider adding numerical data to present the main findings.

Relative to cited references in all text, only 15 (~23%) constitute recent articles, published in last 5 years (2017-2022). Indeed, this point must be reviewed.

In Material and Methods, please, consider presenting levels of physical activity data in the Table 1. Regarding this table, it is necessary describing GXT in the legend.

Moreover, please, consider detailing sports modalities of participants in terms of absolute and relative proportions. How innovative is the current study when compared to Hebisz et al. (2019)?

Additionally, Authors should consider describing a scheme (flowchart) to demonstrate the sequential steps of study and data collection (page 3).

Page 4, second paragraph: the citation “Lazic et al. (2017)” was not found in the References section.

Based on methods of statistical analysis, it is described that the Shapiro-Wilk test was used to assess the distribution characteristics of quantitative results. Considering Table 2, did T1, T2, and T3 results have normal distribution? Indeed, these measures showed high coefficients of variation (%).

Reference

Hebisz R, Hebisz P, Borkowski J, Wierzbicka-Damska I, Zatoń M. Relationship between the skin surface temperature changes during sprint interval testing protocol and the aerobic capacity in well-trained cyclists. Physiol Res. 2019, 68(6), 981-989. doi: 10.33549/physiolres.934114.

Author Response

General comment for Reviewer 2: Thank you for the review comments. We read it in detail and we were trying to make all improvements with the best effort.

As the process of changing the original text was complex, all inserted new text, parts of the text, new words or parts of words were marked in red letters. Corrections are marked in green, as suggested by the reviewer. In light blue and yellow, corrections as suggested by other reviewers.

In the current manuscript, the Authors aimed to investigate whether it is possible to use infrared 8 thermography in the assessment of cardiovascular fitness and aerobic capacity. Although the current issue is interesting, some points are addressed to the Authors, and they need be clarified to improve the manuscript’s quality.

# Considering Abstract, in general, this section is well written. Some few points need to be improved; in Results, please, consider adding numerical data to present the main findings.

Response 1. As suggested by the reviewer, the correlation values between T3 and VO2max and between T3 and Pmax were supplemented.

# Relative to cited references in all text, only 15 (~23%) constitute recent articles, published in last 5 years (2017-2022). Indeed, this point must be reviewed.

Response 2. We do accept reviewer's opinion. Eight older references were replaced by newer ones (since 2017).

# In Material and Methods, please, consider presenting levels of physical activity data in the Table 1. Regarding this table, it is necessary describing GXT in the legend. Moreover, please, consider detailing sports modalities of participants in terms of absolute and relative proportions.

Response 3. The physical activity level of research participants is detailed in section 2.1. Participants

The participants were free of any know neuromusculoskeletal, cardiovascular and respiratory systems impairment. Six of the participants led a sedentary lifestyle, 10 were classified as a physically active (exercise duration 3 to 5 hours per week: 5 – runners, 3 – cross-fit, 2 - swimming, 1 – rocket games;), 7 were classified as athletes (regularly participating in sport competitions; exercise duration 9 to 15 hours per week: 3 – runners, 2 – cross-country skiing, 2 – team games).

The abbreviation GXT has been removed and the full name of the test was wright in the legend.

# How innovative is the current study when compared to Hebisz et al. (2019)?

Response 4. In previous studies, we used a different exercise protocol in which we recorded changes in body surface temperature. At that time, the exercise protocol consisted of 4 sprints, each lasting 30 seconds and separated by a break of 90 seconds. However, such an effort protocol is difficult for non-professional athletes to perform, as we experienced during our research (subjects refused to continue the study, there were some vomiting). Therefore, we have made an attempt to find a protocol based on intense effort, but possible to be performed by healthy people who are not athletes. The exercise protocol, proposed in the reviewed article, turned out to be easier to perform - all participants of the study performed the assigned exercise tests. The above information was included in the abbreviated version in the "Introduction" chapter. In addition, the research described in the reviewed article was enriched (compared to the work from 2019) with the analysis of the restitution of the frequency of heart contractions, as an additional measure of the cardiovascular efficiency. In the previous article, the temperature was measured at two points: the arm and the temple. Only the temperature measured on the temples showed a significant correlation with aerobic capacity, so in the present study, the temperature was measured only from the skin of the temples.

Reference

Hebisz R, Hebisz P, Borkowski J, Wierzbicka-Damska I, ZatoÅ„ M. Relationship between the skin surface temperature changes during sprint interval testing protocol and the aerobic capacity in well-trained cyclists. Physiol Res2019, 68(6), 981-989. doi: 10.33549/physiolres.934114.

# Additionally, Authors should consider describing a scheme (flowchart) to demonstrate the sequential steps of study and data collection (page 3).

Response 5. With the reviewer's suggestion, we created such a scheme. 

# Page 4, second paragraph: the citation “Lazic et al. (2017)” was not found in the References section.

Response 6. This reference was corrected in the text “Suzic Lazic et al. (2017) [6].”

# Based on methods of statistical analysis, it is described that the Shapiro-Wilk test was used to assess the distribution characteristics of quantitative results. Considering Table 2, did T1, T2, and T3 results have a normal distribution? Indeed, these measures showed high coefficients of variation (%).

Response 7. I have sent the individual results of T1, T2, T3, and the result of the Shapiro-Wilk test shows no significant differences compared to the normal distribution in the attachment.

08.06.2022       Corresponding author

Reviewer 3 Report

Critique of “Temporal skin temperature as an indicator of cardiorespiratory fitness assessed with selected methods.”

GENERAL COMMENTS

There is text within the Introduction that pertains to blood distribution, vasoconstriction, and cutaneous blood flow. They change dramatically as a person goes from rest to exercise. Within that text, clearly state when you refer to blood distribution, vasoconstriction, and cutaneous blood flow at rest, and do the same when referring to those same things during exercise.

Lines 44-50: While cutaneous blood flow is low at rest, when one transitions from rest to exercise blood flow, as well as circulating norepinephrine levels, rise. Cutaneous blood flow increases are driven by the greater heat production that occurs with exercise. Norepinephrine by itself may act as a vasoconstrictor, but when its circulating levels rise with concurrent body heat increases, it evokes vasodilation of arterio-venous anastomoses, leading to greater heat losses through the hand’s palmar surface (Caruso et al. IJSM 2015, Grahn et al. JSCR 2012, Kwon et al. JSCR 2013, O’Brien et al. JSCR 2021). Please reexamine these statements and, once again, clarify such statements as occurring during rest or exercise.

Lines 96-99: Briefly describe why subjects performed an incremental test. Presumably it was to establish the exercise intensity of the subsequent test.

Methods: Was the thermal imaging camera used in accordance with the manufacturer's guidelines?

Methods: Prior research (Ogoh & Ainslie JAP 2009) saw cerebral blood flow decline when exercise intensity exceeds 60% VO2 max so the brain is not exposed to excess heat. How do you think this may have impacted your data collection? Please explain.

Methods: What is the relevance of RR (time interval between heart beats)? Wouldn’t HR measurements be a simpler and more direct measurement? Please explain.

Throughout the manuscript change “worm-up” to “warm-up”

SPECIFIC COMMENTS

Line 9: change “in the assessment of” to “to assess”

Line 10: change “in” to “from”

Line 14: change “the athletes” to “athletic”

Line 23: omit the third “the”

Line 24: omit the first “parameters”

Line 27: omit “that”

Line 39: omit “in the body”; change “redistribution” to “distribution”

Line 40: omit “vessels in” and “to the”

Line 42: omit “need” and “get” and “there is”; insert “themselves” after “rid”

Line 43: change “associated with” to “accompany”; omit “the”

Line 49: change “in the” to “to”

Line 50: Insert “The” before “Mentioned”

Line 54: change “which is supposed” to “intended”

Line 56: change “increases in” to “increased”

Lines 56-57: omit “in adaptations to”

Line 57: change “has the effect of lowering” to “lowers”

Line 63: omit “that”

Line 64: omit “the”

Lines 64-65: omit “from a series of 4 sprints such as the Wingate test”

Line 67: change “considered that the” to “surmised those”

Line 67: omit “obtained”; change “size” to “degree”

Line 71: change “own” to “aforementioned” and omit “mentioned above”

Line 72: omit “a break of” and insert “rest periods” after “90 seconds”

Line 74: insert “For these reasons” before “There”; omit “because they anticipate”

Line 75: omit “feeling bad” and “to the sprinting protocol”

Line 92: replace “research” with “study”

Line 115: replace “and breath” with “as their expired”

Line 120: change “using” to “with”

Line 162: make “10-minutes” singular

Line 164: insert “were” after “values”

Line 166: change “measurement has a” to “measurements have”

Line 275: omit “have”

Line 277: omit “performance capacity”

Line 287: place a comma after “sympathetic” and “parasympathetic”; omit “nervous system activity”

Line 309: omit “also”

Author Response

Response to Reviewer 3 Comments

General comment for Reviewer 1: Thank you for the review comments. We read it in detail and we were trying to make all improvements with the best effort.

As the process of changing the original text was complex, all inserted new text, parts of the text, new words or parts of words were marked in red letters. Corrections are marked in yellow, as suggested by the reviewer. In green and yellow, corrections as suggested by other reviewers.

Comments and Suggestions for Authors

Critique of “Temporal skin temperature as an indicator of cardiorespiratory fitness assessed with selected methods.”

GENERAL COMMENTS

# There is text within the Introduction that pertains to blood distribution, vasoconstriction, and cutaneous blood flow. They change dramatically as a person goes from rest to exercise. Within that text, clearly state when you refer to blood distribution, vasoconstriction, and cutaneous blood flow at rest, and do the same when referring to those same things during exercise.

Response 1. We are agreeing with the reviewer, and we pointed out in paragraph 2 when we refer to blood distribution, vasoconstriction and cutaneous blood flow during the exercise.

# Lines 44-50: While cutaneous blood flow is low at rest, when one transitions from rest to exercise blood flow, as well as circulating norepinephrine levels, rise. Cutaneous blood flow increases are driven by the greater heat production that occurs with exercise. Norepinephrine by itself may act as a vasoconstrictor, but when its circulating levels rise with concurrent body heat increases, it evokes vasodilation of arterio-venous anastomoses, leading to greater heat losses through the hand’s palmar surface (Caruso et al. IJSM 2015, Grahn et al. JSCR 2012, Kwon et al. JSCR 2013, O’Brien et al. JSCR 2021). Please reexamine these statements and, once again, clarify such statements as occurring during rest or exercise.

Response 2. For better clarity, we reworded indicated by the reviewer text. Corrections made are marked in yellow.

During intense exercise, the concentration of norepinephrine in a blood increases [16,17]. Norepinephrine acts on the postsynaptic receptors (alpha1, alpha2) of the sympathetic adrenergic system, causing vasoconstriction of the cutaneous circulation vessels [11,18]. As a result, at the onset of intense exercise, there are changes in blood flow, consisting of an increase in flow in the limbs loaded with exercise and a decrease in flow to inactive limbs and skin microcirculation [18,19,20]. Through the described reactions, the muscles are better supplied with oxygen during intense work and the convection of flowing blood is more effective [12-15,21]. However, when exercise is performed over a long period of time, the internal body temperature increases and the heat removal mechanisms must be activated [22]. These include the vasodilation of the skin vessels and an increase in the intensity of cutaneous blood flow, intended to increase the heat release to the atmosphere [22]. The magnitude of the increase in cutaneous blood flow during exercise depends on aerobic capacity [11] and increased adaptation to training and heat stress [23]. This adaptation lowers the internal temperature threshold at which vasodilation of the skin vessels occurs [23]. Exercise-induced changes in blood flow affect body surface temperature [20, 24].

# Lines 96-99: Briefly describe why subjects performed an incremental test. Presumably it was to establish the exercise intensity of the subsequent test.

Response 3. With the Reviewer's suggestion, this information was supplemented.

”The subjects had not performed heavy physical exercise in the 48 hours prior to the exercise tests. Each subject performed an incremental test for cardiovascular fitness and aerobic efficiency assessment as well as to determine the power of verification test equal to 110% of Pmax obtained in GXT”

# Methods: Was the thermal imaging camera used in accordance with the manufacturer's guidelines?

Response 4. Yes, the relevant statement was included in the text (page 4; paragraph 3; line 3 – highlighted in blue)

# Methods: Prior research (Ogoh & Ainslie JAP 2009) saw cerebral blood flow decline when exercise intensity exceeds 60% VO2 max so the brain is not exposed to excess heat. How do you think this may have impacted your data collection? Please explain.

Response 5.

  • Ogoh found that the vascular bed in the cerebral circulation is small. Therefore, changes in blood flow at an intensity greater than 60% of VO2max should have little impact on our results. We believe that our results are more dependent on the exercise and restitution intensity of extracranial circulation.
  • Ogoh claims that a decrease in the intensity of cerebral circulation occurs at the intensity of 60% VO2max due to hyperventilation and a decrease in blood PaCO2. The increase in ventilation on the scale expressed in %VO2max depends on efficiency. For this reason, the ventilation threshold occurs at a higher %VO2max among athletes than non-athletes. As a result, it can be expected that a decrease in the intensity of cerebral blood flow will occur at a higher intensity in training (efficient) people compared to non-training people. If it has an impact on the results obtained by us, then among the athletes (efficient) this impact should be less prominent than among the non-training.
  • It is difficult to assess the effect of reducing cerebral blood flow at an intensity exceeding 60%VO2max. Does it reduce the temperature of the surface of the temples, because the tissues located inside the skull heat up less and then the conduction between the surface of the temples and the tissues located deeper is less intense? Maybe it increases the temperature of the temples, because lowering the cerebral blood flow may allow blood to be redistributed to other tissues (including skin circulation)?
  • Currently, we are planning to perform further tests in which efforts will be performed at an intensity of 2W / kg, 3W / kg, 4W / kg, 5W / kg body weight. If a decrease in the intensity of cerebral blood flow affects changes in the temperature of the surface of the temples during exercise and in restitution, then perhaps the results of the planned research will allow this effect to be noticed. It can lead to different thermal responses to low-intensity (2W/kg) and high-intensity (4W/kg, 5W/kg) effort. We are interested in your participation in the interpretation of the data we obtain.

# Methods: What is the relevance of RR (time interval between heart beats)? Wouldn’t HR measurements be a simpler and more direct measurement? Please explain.

Response 6. Indeed, both of these parameters (RR and HR) showed a relationship with T3 and one could limit ourselves to showing only the relationship with HR in the paper, which, as the reviewer pointed out, is an easier and directly measurable parameter. On the other hand, it is HRV that is a direct exponent of the sympathetic-parasympathetic balance, which depends, inter alia, on thermal stress. Keeping both parameters in the presented work gives a more complete picture of the relationship between the post-exercise thermoregulation process and cardiovascular capacity. Additionally, it can also increase the audience.

# Throughout the manuscript change “worm-up” to “warm-up”

Response 7. Thank you for this comment. That was corrected throughout the text.

# SPECIFIC COMMENTS

Response 8. In accordance with the reviewer's suggestions, all corrections have been made.

Line 9: change “in the assessment of” to “to assess”

Line 10: change “in” to “from”

Line 14: change “the athletes” to “athletic”

Line 23: omit the third “the”

Line 24: omit the first “parameters”

Line 27: omit “that”

Line 39: omit “in the body”; change “redistribution” to “distribution”

Line 40: omit “vessels in” and “to the”

Line 42: omit “need” and “get” and “there is”; insert “themselves” after “rid”

Line 43: change “associated with” to “accompany”; omit “the”

Line 49: change “in the” to “to”

Line 50: Insert “The” before “Mentioned”

Line 54: change “which is supposed” to “intended”

Line 56: change “increases in” to “increased”

Lines 56-57: omit “in adaptations to”

Line 57: change “has the effect of lowering” to “lowers”

Line 63: omit “that”

Line 64: omit “the”

Lines 64-65: omit “from a series of 4 sprints such as the Wingate test”

Line 67: change “considered that the” to “surmised those”

Line 67: omit “obtained”; change “size” to “degree”

Line 71: change “own” to “aforementioned” and omit “mentioned above”

Line 72: omit “a break of” and insert “rest periods” after “90 seconds”

Line 74: insert “For these reasons” before “There”; omit “because they anticipate”

Line 75: omit “feeling bad” and “to the sprinting protocol”

Line 92: replace “research” with “study”

Line 115: replace “and breath” with “as their expired”

Line 120: change “using” to “with”

Line 162: make “10-minutes” singular

Line 164: insert “were” after “values”

Line 166: change “measurement has a” to “measurements have”

Line 275: omit “have”

Line 277: omit “performance capacity”

Line 287: place a comma after “sympathetic” and “parasympathetic”; omit “nervous system activity”

Line 309: omit “also”

08.06.2022       Corresponding author

Round 2

Reviewer 1 Report

The authors have correctly made the changes suggested by the reviewer and the quality of the article has improved significantly.

No further revisions by the authors are required. 

Author Response

We would like to thank you for the time and effort spent reviewing our manuscript and outstanding support for its improvement. 

Some additional text, as well as corrections, was added in accordance with suggestions given by one of the Reviewers. The brand new text and new corrections are highlighted in grey.

As suggested by Reviewer 

#1 in the Results chapter (page 5, paragraph 2), we explained why we used RR intervals and not the simpler HR measurement;

"...The RR intervals allowed for the analysis of consciously selected sections for HRR analysis. A simpler method such as HR recording would result in a random entry segment for HRR analysis as the "cardiofrequencymeter's" software calculates the HR based on the averaging data at 3 second intervals. The conversion of RR to HR was performed automatically by PolarFlow (www.flow.polar.com) in a precisely marked part of saved data. ..."

#2 in the discussion chapter (page 10, paragraph 2), we referred to the suggested publication, trying to explain how the results presented in it (increased cerebral blood flow in efforts performed with intensity above 60% of VO2max) may impact the data obtained by us.

"The results of the research presented in this paper indicate that the temperature of the temples' surface increases in recovery after intense aerobic exercise, especially in people with high physical capacity. Such results confirm the thesis from the studies by Hebisz et al. (25) that blood flow increases in the temporal region as a result of intensive work. At the same time, already at an intensity above 60% VO2max, the cerebral blood flow decreases, which protects the brain from excessive thermal stress (68). It is possible that the reduction in cerebral blood flow additionally increases the blood flow in the branches of the external carotid artery. Thus, a decrease in cerebral blood flow may result in a greater increase in temples surface temperature following exercise. However, this statement requires empirical confirmation."

19.06.2022  Corresponding author

Reviewer 2 Report

In response to major review, the manuscript was improved and is suitable to being published.

Author Response

(The authors gave the same response as above.)

Reviewer 3 Report

Critique of “Temporal skin temperature as an indicator of cardiorespiratory fitness assessed with selected methods.”

GENERAL COMMENTS

Methods: Prior research (Ogoh & Ainslie JAP 2009) saw cerebral blood flow decline when exercise intensity exceeds 60% VO2 max so the brain is not exposed to excess heat. How do you think this may have impacted your data collection? Please explain. 

Methods: What is the relevance of RR (time interval between heart beats)? Wouldn’t HR measurements be a simpler and more direct measurement? Please explain. 

SPECIFIC COMMENTS

Line 10: change “in” to “from”

Line 40: change “in” to “to”

Line 83: insert “It is” before “Surmised”

Line 84: change “area” to “region”

Line 88: should be reworded to read as “90 second rest periods”

Line 90: should be reworded to read as “For these reasons”

Line 90: should be reworded to read as “[28]. An”

Line 107: change “rocket” to “racket”

Line 261: insert “in accordance” before “with”; change “manufactorer’s” to “manufacturer’s”

Line 272: change “were” to “was”

Line 410: change “presented” to “present”

Line 440: change “determined” to “reduced”

Line 454: should be reworded to read as “and current results show this correlation…”

Line 455: change “presented” to “the current”

Author Response

Response to Reviewer Comments

We would like to thank you for the time and effort spent reviewing our manuscript and outstanding support for its improvement. We found your comments and suggestions very helpful and strongly believe they improved the quality of this manuscript. Your original comments and our responses to those comments are listed below.

Comments and Suggestions for Authors

Critique of “Temporal skin temperature as an indicator of cardiorespiratory fitness assessed with selected methods.”

GENERAL COMMENTS

Methods: Prior research (Ogoh & Ainslie JAP 2009) saw cerebral blood flow decline when exercise intensity exceeds 60% VO2 max so the brain is not exposed to excess heat. How do you think this may have impacted your data collection? Please explain. 

Answer to the reviewer: Thank you for this constructive comment. We are sorry for omitting this throughout the manuscript. We have referred to the aforementioned research paper in the discussion of the revised manuscript: page 10; paragraph 2; the new text is highlighted in grey. The references have also been supplemented with the suggested position.

The new paragraph:

“The results of the research presented in this paper indicate that the temperature of the temples' surface increases in recovery after intense aerobic exercise, especially in people with high physical capacity. Such results confirm the thesis from the studies by Hebisz et al. (25) that blood flow increases in the temporal region as a result of intensive work. At the same time, already at an intensity above 60% VO2max, the cerebral blood flow decreases, which protects the brain from excessive thermal stress (68). It is possible that the reduction in cerebral blood flow additionally increases the blood flow in the branches of the external carotid artery. Thus, a decrease in cerebral blood flow may result in a greater increase in temples surface temperature following exercise. However, this statement requires empirical confirmation.”

Methods: What is the relevance of RR (time interval between heart beats)? Wouldn’t HR measurements be a simpler and more direct measurement? Please explain. 

Answer to Reviewer:  Indeed, the use of the HR records would be simpler, but it would not be more direct. This is because each 'cardiofrequencymeter' measures the time interval between heart beats (RR). The cardiofrequencymeter's software calculates HR values based on the recorded RR. Most often, HR is calculated on the basis of data obtained from averaging at 3-second intervals. Such a method of recording makes the beginning and end of the segment selected for analysis random. Our method of analysis allowed for a conscious selection of sections for HRR analysis, which made the analysis very accurate. I would like to add that the conversion of RR to HR is performed automatically by the PolarFlow program in the precisely marked part of the saved data. We are agreeing with the reviewer, that such an explanation should appear in the main text, and we do include brand new text in the Results chapter. The added text can be found on page 5; paragraph 2; the new text is highlighted in grey.

“During the warm-up and a 10-minute passive break before the 110%Pmax power test, the time interval between heartbeats (RR) was recorded using a V800 cardiofrequency meter (Polar, Oy, Finland). Heart rate values were averaged for 59-61" (HRR1'), 119-121" (HRR2'), 179-181" (HRR3'), 239-241" (HRR4') and 299-301" (HRR5') recovery after warm-up, as recovery RR interval measurements have a high variability. The RR intervals allowed for the analysis of consciously selected sections for HRR analysis. A simpler method such as HR recording would result in a random entry segment for HRR analysis as the "cardiofrequencymeter's" software calculates the HR based on the averaging data at 3 second intervals. The conversion of RR to HR was performed automatically by PolarFlow (www.flow.polar.com) in a precisely marked part of saved data. The changes in heart rate during recovery after warm-up were then calculated as the difference between the heart rate measured at the end of warm-up and HRR1' (ΔHRR1'), HHR2' (ΔHRR2'), HRR3' (ΔHRR3'), HRR4 (ΔHRR4') and HRR5' (ΔHRR5'), respectively, similar to Suzic Lazic et al. (2017) [6]. “

SPECIFIC COMMENTS

Answer to Reviewer: In accordance with the reviewer's suggestions, all corrections have been made. They are highlighted in grey.

Line 10: change “in” to “from”

Line 40: change “in” to “to”

Line 83: insert “It is” before “Surmised”

Line 84: change “area” to “region”

Line 88: should be reworded to read as “90 second rest periods”

Line 90: should be reworded to read as “For these reasons”

Line 90: should be reworded to read as “[28]. An”

Line 107: change “rocket” to “racket”

Line 261: insert “in accordance” before “with”; change “manufactorer’s” to “manufacturer’s”

Line 272: change “were” to “was”

Line 410: change “presented” to “present”

Line 440: change “determined” to “reduced”

Line 454: should be reworded to read as “and current results show this correlation…”

Line 455: change “presented” to “the current”

19.06.2022      Corresponding author